# Impact of Short-Acting Disopyramide on Left Ventricular Mechanics Evaluated by Strain Analysis in Patients with Hypertrophic Obstructive Cardiomyopathy

**DOI:** 10.3390/jcm11247325

**Published:** 2022-12-09

**Authors:** Idit Yedidya, Gabby Elbaz Greener, Mordehay Vaturi, Alik Sagie, Offer Amir, Shemy Carasso, Daniel Monakier

**Affiliations:** 1Department of Cardiology, Rabin Medical Center, Beilinson Hospital, Petach Tikva 4941492, Israel; 2Sackler Faculty of Medicine, Tel Aviv University, Tel Aviv 6329302, Israel; 3Department of Cardiology, Hadassah Medical Center, The Faculty of Medicine, Hebrew University of Jerusalem, Jerusalem 91120, Israel; 4Baruch Padeh Medical Center, Poriya 15208, Israel; 5Azrieli Faculty of Medicine in the Galilee, Bar Ilan University, Safed 1311502, Israel

**Keywords:** echocardiography, hypertrophic obstructive cardiomyopathy, HOCM, disopyramide, strain

## Abstract

Background: Disopyramide is a class Ia antiarrhythmic drug that has been used for the second-line treatment of symptomatic hypertrophic obstructive cardiomyopathy (HOCM). The aim of the study was to assess the impact of short-acting disopyramide in patients with hypertrophic obstructive cardiomyopathy (HOCM) using two-dimensional speckle-tracking echocardiography. Methods: This prospective study included patients with HOCM on chronic treatment with short-acting disopyramide. Two sequential comprehensive echocardiographic examinations were performed: after temporary disopyramide suspension and 2.5 h after disopyramide intake. Results: 19 patients were included in the study. The effect of disopyramide on the left ventricle was not uniform. After the intake of disopyramide, the mean global strain peak was −17 ± 2% before disopyramide intake and −14 ± 2% after (*p* < 0.0001). There was a significant reduction in strain in the basal septal (*p* = 0.015), basal inferior (*p* = 0.019), basal posterior (*p* = 0.05), apical anterior (*p* = 0.0001), and apical lateral segments (*p* = 0.021). In all other segments, there was no significant change. Disopyramide also caused a significant accentuation of the base-apex strain gradients (*p* = 0.036). No change was noted in circumferential and left atrial strain. While the left ventricular ejection fraction and outflow gradients did not change, the significant reduction in global and segmental longitudinal strain demonstrated the acute negative inotropic effect of disopyramide on the myocardium in patients with HOCM. Conclusion: A strain analysis may be a useful tool to assess the negative inotropic effect of cardiovascular medication on the left ventricle in patients with HOCM.

## 1. Introduction

Disopyramide is a class Ia antiarrhythmic drug that has been used since the early 1980s for the second-line treatment of symptomatic hypertrophic obstructive cardiomyopathy (HOCM) [1,2]. A negative inotropic effect of the drug has been thought to account for the reduction in the left ventricular (LV) outflow tract (LVOT) gradient. However, support for this theory has been inconsistent. Some studies have failed to demonstrate a negative inotropic effect of the drug, and in others, the effect was restricted only to patients with LV dysfunction [3,4,5,6,7,8,9,10]. These discrepancies may be at least partly explained by the small and nonuniform populations evaluated in several studies, including patients with and without LV dysfunction, and differences in the protocols of disopyramide administration among the studies.

Myocardial mechanics evaluations have shown that the decrease in the LVOT gradient during septal ethanol ablation coincided with global LV dysfunction despite the presence of only local ischemia during septal balloon occlusion. The global dysfunction was transient and the gradient partially rebounded when global function improved and dysfunction was limited to the basal septum [11].

Prompted by these findings, our study aimed to evaluate the acute effect of disopyramide treatment in patients with HOCM using speckle echocardiography, integrating myocardial strain analysis with two-dimensional imaging.

## 2. Methods

### 2.1. Study Population

Patients were recruited prospectively at the Hypertrophic Cardiomyopathy Clinic of Rabin Medical Center from November 2016 to December 2017. Inclusion criteria were HOCM, chronic treatment with short-acting disopyramide, and normal LV systolic function (LV ejection fraction ≥ 50%). Patients who had prosthetic valves or pacemakers were excluded. Patients with atrial fibrillation at the time of echocardiography were also excluded. All patients included in the study provided written informed consent to participate.

### 2.2. Study Protocol

Each patient underwent two sequential comprehensive echocardiography examinations on the same day, before and after disopyramide intake. The first study was performed at 8:30 AM, at least 12 h after intake of the last usual dose of disopyramide (Disopyramide, Trima Pharmaceuticlas, Maabarot, Israel). Patients were instructed not to otherwise change their regular medication regimen. They were told to avoid coffee or other stimulating drinks and to estimate the amount of fluid intake prior to the test. Immediately upon completion of the first echocardiographic examination, patients were asked to take their usual morning dose of disopyramide in the presence of the investigators. The second test was performed at least 2.5 h after the first.

The study protocol was approved by the local institutional clinical research board RMC-538-16 and registered in the NIH as NCT02917395.

### 2.3. Echocardiographic Examinations

All echocardiographic examinations were performed with a single machine (Epic7 Version 1.7.1, Phillips Ultrasound, Inc., Bothell, WA, USA) using an S5-1 probe. Studies included parasternal long-axis views, three levels of short-axis views (mitral, midpapillary, and apical levels), and three apical long-axis views (each repeated to include both the left atrium and ventricle with decreased depth to focus on the left ventricle). All cine acquisitions were spanned over two or three consecutive beats. The echocardiographic studies were interpreted at the time of acquisition. Two-dimensional Doppler parameters were measured according to the guidelines of the American Society of Echocardiography [12,13].

### 2.4. Strain Analysis

Strain measurements were performed using two-dimensional tissue-tracking software (Velocity Vector Imaging; Siemens Medical Solutions USA Inc., VVI, Malvern, PA, USA) from archived two-dimensional echocardiographic studies. Longitudinal wall strain and strain rate were averaged from 18-segment measurements from the apical two-, three-, and four-chamber views. Circumferential strain, strain rate, and rotation angles were measured in six segments per short-axis plane at the parasternal mid-LV level and the apical level obtained in a window between the parasternal and apical windows. All measurements were averaged for each short-axis level. Peak systolic rotation was defined as the peak angular short-axis myocardial displacement during systole. Peak myocardial systolic rotation angles were used to calculate midapical LV twist, defined as the maximal instantaneous mid- to apical rotation angle difference. All myocardial mechanical analyses were automatically averaged on all consecutive beats available on stored cine clips. VVI strain analysis was not expected to depend on source acquisition hardware. Strain analysis was performed by a single echocardiographer (S.C.) blinded to the clinical data, standard echocardiography results, and timing.

### 2.5. Statistical Analysis

The data were analyzed using MedCalc version 11.6.1 (MedCalc Software, Mariakerke, Belgium). Continuous data are reported as mean ± SD. Sample size assessments were calculated to detect a significant drop in longitudinal strain from 17% ± 2.5 (SD) (absolute value, typical of HCM patients) of 3% (to 14%) with the lowest probability type I (*p* < 0.05) and II (*p* < 0.01) errors and was found to be 14 patients. We prospectively recruited 35% more patients for the study. To ascertain the normal distribution of pre- and postdisopyramide strain measurements, the Shapiro-Wilk test was used. After confirmation of normality, we assessed the differences between pre- and post-administration of disopyramide and were compared by the parametric paired, two-tailed *t*-test. Statistical significance was determined with *p* < 0.05.

## 3. Results

Twenty-one patients met the inclusion criteria and consented to participate in the study. Two of them were later excluded because of atrial fibrillation and unsatisfactory imaging quality. The mean age of the remaining 19 patients was 66.1 ± 9.45 years; 57% were men. The mean time from diagnosis of HOCM was 8.0 ± 9.9 years. The average daily dose of disopyramide was 380 ± 150 mg. Most (13, 68%) had dyspnea or angina (3, 16%), and were in 2 ± 1 NYHA class.

### 3.1. Conventional Echocardiography

The first echocardiographic examination demonstrated normal left ventricular systolic function by volume measurements (using Simpson’s rule) with low tissue Doppler longitudinal systolic velocities (Table 1). The right ventricular longitudinal systolic function was within normal range by both tissue Doppler and tricuspid annular plane systolic excursion (TAPSE). Left atrial volumes were mild to moderately enlarged, with mitral E/e’ suggesting the presence of grade 2 LV diastolic dysfunction. All patients had evidence of LVOT obstruction with mitral systolic anterior motion, with a gradient as high as 43 ± 28 mmHg at rest, increasing by ~50% with provocation (Valsalva maneuver). None of the conventional echocardiographic parameters changed at 150 min after administration of disopyramide, including LVOT gradients.

### 3.2. Systolic Mechanical Imaging

The biplane VVI-measured LV ejection fraction was lower than calculated conventionally and decreased after the administration of disopyramide (Table 2). Figure 1 demonstrates the results of the Shapiro-Wilk test confirming the normality of the distribution of strain measurements. The predisopyramide global longitudinal strain was lower than normal (absolute shortening value at aortic valve closure time), and lowest at the septal and basal segments. After disopyramide administration, the global longitudinal strain further decreased. Systolic strain decreased in nearly all segments, however, the difference was statistically significant only in some, mainly in the basal and apical regions, with no specific pattern of distribution (Table 2). The circumferential systolic strain was normal and remained unchanged after disopyramide. Peak systolic apical rotation and left ventricular twist (difference in apical-to-basal rotation angle) were supranormal before disopyramide and significantly decreased after.

### 3.3. Diastolic Mechanical Imaging (Reverse Rotation and Atrial Function)

The fraction of early reversed apical rotation (fraction change from peak rotation angle to rotation angle at 10% cycle length later in diastole) was in the normal range, suggesting normal left ventricle relaxation before and after disopyramide. Left atrial function by strain and phasic volumes also followed a normal pattern with a normal reservoir, passive function, and active function. The atrial function did not change after the intake of disopyramide.

### 3.4. Interobserver Variability

Interobserver variability, calculated in nine randomly selected patients from the study population, was 8.4% for the first echocardiographic examination and 1.6% for the second examination.

## 4. Discussion

This prospective study showed that short-acting disopyramide significantly reduced global segmental longitudinal strain. To our knowledge, this is the first study to evaluate the acute effect of disopyramide on myocardial mechanics as measured with strain in patients with HOCM.

Our baseline findings of good systolic left ventricular and diastolic dysfunction are characteristic of patients with HOCM. The strain evaluation at baseline showed reduced values of global longitudinal strain and, to a lesser extent, of circumferential strain, in agreement with previous reports on HOCM [14]. After the administration of disopyramide, there was a further significant reduction in systolic global and segmental longitudinal and rotational mechanics. These findings seem consistent with the negative inotropic properties of disopyramide. The decrease in longitudinal strain affected nearly all segments but was significant only in some, mostly in segments where baseline strain was higher. An abnormally low baseline value with a further decrease in strain was demonstrated only in the basal septal segment, which is also the thickest. This unique finding is probably directly related to the beneficial effect of disopyramide in the relief of LVOT obstruction in patients with HOCM. In previous studies of nonobstructive hypertrophic cardiomyopathy (HCM), LV function was evaluated by strain mechanics regardless of medical treatment. One study reported low septal strain in asymmetrical HCM [15] that was correlated to the septum and posterior wall thickness ratio [16]. Another noted a correlation between a segmental strain with wall thickness [17]. In a study measuring strain before, during, and after septal alcohol ablation [11] alcohol was found to acutely reduce global left ventricular strain. GLS was correlated with the LVOT gradient, showing that a decrease of LVOT gradients to less than 30 mmHg would require GLS to be less (negative) than −11%. This reduction was not reached in our current study, maybe because the postadministration echo was done before the needed 3–4 h of plasma peak levels of disopyramide.

In a comprehensive study of patients with HOCM undergoing surgical myectomy, the reduced longitudinal strain was found to correlate with disarray and fibrosis rather than with septal thickness [18]. We could not assess the association between fibrosis and the effect of disopyramide as only one patient underwent a cardiac magnetic resonance (CMR) study to assess fibrosis.

A study evaluating LV function after septal ablation using CMR strain analysis showed a remote effect with an improvement in lateral wall function after septal ablation [19]. Accordingly, disopyramide may have a remote segmental effect as well as a negative inotropic effect, as suggested by the segmental pattern of changes found in the present cohort. Alternatively, the remote effect may be due to the relief of obstruction per se.

We did not find an effect of disopyramide on global synchrony. Therefore, the negative inotropic effect is apparently not mediated by changes in the LV cycle.

A significant apex-to-base decrease in longitudinal strain was reported in a previous study of 39 patients with HCM analyzed by the presence/absence of myocardial fibrosis on CMR [20]. Except in three patients with apical hypertrophy, myocardial fibrosis was associated with depressed longitudinal strain. In the present cohort, we too observed a similar pattern of a lower apex-to-base longitudinal strain gradient, which was accentuated in the second examination, as well as a significant base-to-apex gradient (*p* = 0.03).

In our study, the circumferential strain was normal, and the rotation and twist were higher than normal. No change occurred in circumferential strain after the administration of disopyramide, although apical systolic rotation and twist decreased (Table 2). The drug did not seem to have any diastolic effects as assessed by the fraction of early reversed apical rotation and phasic left atrial volumes and function.

Short-acting disopyramide had no effect on other echocardiographic parameters, such as the ejection fraction or LVOT gradients. This finding is probably attributable to the short wash-out period of the medication in our chronically disopyramide-treated patients. Moreover, it may have been the dose of the drug that may have been too low to be effective on the LVOT gradient [21]. Yet the significant systolic effect of the drug on the local and global longitudinal strain and rotation was clearly demonstrated. The significant findings on evaluation by strain imply that the action of disopyramide is not restricted to its negative inotropic effect but may also involve other mechanisms such as a regional effect on the basal/middle segments and a change in both the LVOT and the intraventricular gradients.

As for limitations of the study, in accordance with the instructions of the hospital’s clinical research board, we stopped disopyramide for 12 h before the first echocardiographic examination. The brevity of the washout period may have led to an underestimation of the full effect of the drug, although it was long enough to yield significant changes in strain between the two echocardiographic examinations. Similarly, the dysopiramide dose may have been too low to exert its full effect on the LVOT gradient.

Nearly all patients were on chronic beta-blocker therapy, so the echocardiographic parameters in our study reflect a beta-blockade effect present already at the baseline examination. Nevertheless, as the beta blockers were long acting and were administered in a daily dose, there was no significant difference in blood pressure or heart rate between the two echocardiographic examinations. We, therefore, believe that the beta blocker treatment did not influence the changes observed in the second echocardiographic examination.

In conclusion, disopyramide significantly reduces global longitudinal strain, segmental longitudinal strain, the base-to-apex gradient, and systolic rotational mechanics. The significant reduction in strain parameters may explain the mechanism of action of disopyramide in HOCM as an acute decrease in the myocardial inotropic state. A strain analysis may be a useful tool to assess the negative inotropic effect of cardiovascular medications on the left ventricle in patients with HOCM.

## Figures and Tables

**Figure 1 jcm-11-07325-f001:**
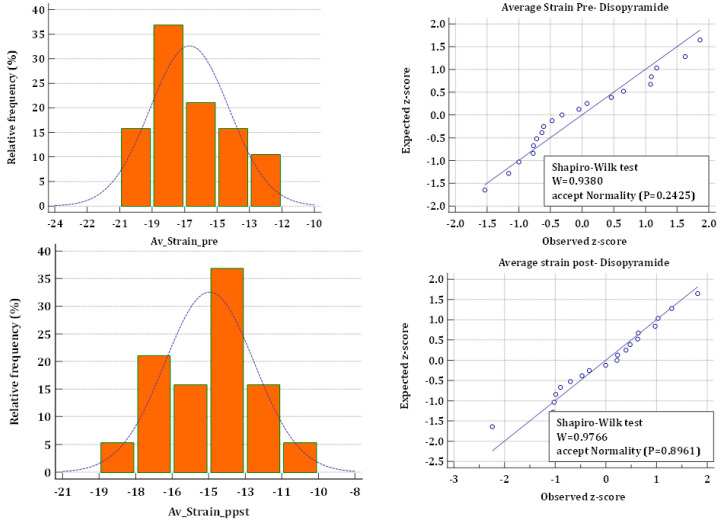
Shapiro-Wilk test for Normality of strain measurements.

**Table 1 jcm-11-07325-t001:** Conventional Echocardiography results in 19 patients with HOCM (mean ± SD).

Measurement	Before Disopyramide	After Disopyramide	*p*-Value
BP dia (mmHg)	74.84 ± 9.2	72.2 ± 10.9	0.1054
BP sys (mmHg)	136.5 ± 19.26	129.84 ± 18.3	0.1419
Pulse (b/min)	62.05 ± 10.88	60.1 ± 8.6	0.8206
**Systolic**			
EF Simpson (%)	71.68 ± 5.4	70.1 ± 4.7	0.1499
Rest gradient mmHg	42.7 ± 28.4	43.47 ± 28.4	0.9742
Gradients during Valsalva (mmHg)	62.1 ± 30.6	62.26 ± 34.63	0.6025
S ant (cm/s)	7 ± 1.2	7.7 ± 2.2	0.9765
S inf (cm/s)	7.7 ± 1.6	7.6 ± 1.7	0.9743
S lat (cm/s)	8.6 ± 1.3	8.3 ± 1.3	0.6522
S sep (cm/s)	6.7 ± 1.5	6.6 ± 1.2	0.9523
RV s (cm/s)	13.46 ± 3.9	12.6 ± 2.9	0.3369
TAPSE (cm)	22.37 ± 3.3	22.68 ± 4.0	0.5898
LA volume (ml)	80.42 ± 23.3	76.17 ± 22.18	0.6165
MR (grade)	2.63 ± 0.8	2.68 ± 1	1
RVSP (mmHg)	17.47 ± 17.6	15.47 ± 17.55	0.1445
**Diastolic**			
E (cm/s)	82.6 ± 20.2	85 ± 22.6	0.2
A (cm/s)	88.7 ± 26.6	83.5 ± 23	0.3114
E/A	1.0 ± 0.42	1.1 ± 0.47	0.2111
Dec time (ms)	232.1 ± 70.4	233 ± 79	0.774
e’ lat (cm/s)	7.1 ± 2	6.6 ± 1.8	0.1011
e’ sep (cm/s)	5.3 ± 1.04	5.33 ± 1.4	0.757
E/e’ lat (cm/s)	12.47 ± 5.3	12.94 ± 5.8	0.1042
E/e’ sep (cm/s)	15.8 ± 4.7	15.9 ± 5.8	0.1688
Pul S (cm/s)	57.3 ± 12.3	54 ±11.7	0.6046
Pul D (cm/s)	52.9 ± 17.2	56.3 ± 18.4	0.2959
e’ ant (cm/s)	6 ± 2	5.8 ± 2	0.587
e’ inf (cm/s)	5.75 ± 1.3	5.78 ± 1.8	0.9239

**Table 2 jcm-11-07325-t002:** Longitudinal and circumferential strain results (%) in 19 patients with HOCM (mean ± SD).

Measurements	Before Disopyramide	After Disopyramide	*p*-Value
**Longitudinal Mechanics**			
**Bi-plane VVI LVEF**	53 ± 6	50 ± 5	0.017
**Global**			
1. Average strain peaks	−17 ± 2	−14 ± 2	<0.0001
2. Global strain at AVC	−15 ± 2	−13 ± 2	<0.0001
Global synchrony (%)	92 ± 5	91 ± 5	0.26
**Segmental**			
06-basal septal	−14 ± 5	−10 ± 4	0.015
12-mid septal	−14 ± 5	−13 ± 5	0.18
16-apical septal	−21 ± 8	−21 ± 6	0.93
14-apical lateral	−17 ± 5	−14 ± 5	0.021
09-mid lateral	−15 ± 6	−13 ± 5	0.086
03-basal lateral	−20 ± 7	−18 ± 5	0.24
05-basal inferior	−16 ± 6	−13 ± 5	0.019
11-mid inferior	−12 ± 6	−10 ± 5	0.31
15-apical inferior	−21 ± 4	−20 ± 5	0.13382
13-apical anterior	−16 ± 5	−11 ± 5	0.00012
08-mid anterior	−16 ± 3	−15 ± 5	0.28
02-basal anterior	−18 ± 5	−16 ± 5	0.13
04-basal posterior	−19 ± 6	−16 ± 4	0.046
10-mid posterior	−14 ± 5	−12 ± 5	0.28
14-apical posterior	−16 ± 6	−15 ± 4	0.85
13-apical anteroseptal	−20 ± 5	−18 ± 5	0.056
07-mid anterosepta	−18 ± 5	−15 ± 6	0.064
01-basal anteroseptal	−14 ± 4	−12 ± 5	0.057
**Circumferential Mechanics**			
Base Circ Strain	−22 ± 11	−24 ± 6	0.38
Mid Circ Strain	−24 ± 5	−28 ± 6	0.97
Apex Circ Strain	−36 ± 8	−37 ± 7	0.98
Global CS at AVC	−28 ± 8	−27 ± 9	0.29
**Rotation (°)**			
Base	−8 ± 3	−6 ± 3	0.094
Mid	1 ± 6	2 ± 5	0.56
Apex	11 ± 4	8 ± 4	0.036
Twist (°)	16 ± 5	12 ± 6	0.001
**FEARR (%)**	37 ± 14	35 ± 18	0.63

VVI, velocity vector imaging; LVEF, left ventricular ejection fraction; AVC, aortic valve closure; CS, circumferential strain, FEARR, fraction of early apical reverse rotation. Global synchrony is calculated as the fraction of global strain at AVC over the average of peaks of segmental strain.

## Data Availability

The data are available upon request from the corresponding author.

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
