# Peer review of "Impact of Short-Acting Disopyramide on Left Ventricular Mechanics Evaluated by Strain Analysis in Patients with Hypertrophic Obstructive Cardiomyopathy"

_jcm, 2022, doi:10.3390/jcm11247325_

Round 1

Reviewer 1 Report (Previous Reviewer 2)

The focus of the present manuscript is the effect of disopyramide on GLS measurements in 19 HOCM patients. The dose of disopyramide is (still) unknown and the time of administration is described as 'approximately 2,5 hours after the first [echo] '. This cannot be a valid description of your methods, and a baseline characteristic should be included of age, sex, and at least disopyramide doses.

Results: The average GLS decreased from -17% to -14% after disopyramide in the dose that was given. In particular, GLS was already lower in some basal and septal parts and after disopyramide the baseline GLS decreases in the basal anteroseptal part from -14 to -12% (p= 0,057).

In the abstract and manuscript this anteroseptal part that would be mainly involved in obstruction is converted in a 'basal septal segment involved in left ventricular outflow tract obstruction' but it was the average basal inferior septal segment in your study that was significantly decreased, and to a lesser extent the anteroseptal basal part.

Possibly, the paired intraindividual results/ changes in anteroseptal GLS would be useful to see, and maybe they correlate with LVOT obstruction. It could be that disopyramide is not useful for everyone, and only in those with higher anteroseptal basal GLS (so that there is something to inhibit)?  

At present, there are discrepancies that would really prevent this paper from being understood. The conclusion that one would be tempted to follow is that despite GLS reductions, this is not enough GLS reduction to cause a significant effect on LVOT gradients, neither was this reduction enough to cause a significant effect on LVEF.  

The real question is how much GLS decrease is necessary for a LVOT gradient decrease and at which disopyramide dose does the LVOT gradient decrease, and this is not answered in the present study, but is only made difficult to understand.    

Author Response

Reviewer 1 Comments

Comment 1:

“The focus of the present manuscript is the effect of disopyramide on GLS measurements in 19 HOCM patients. The dose of disopyramide is (still) unknown and the time of administration is described as 'approximately 2,5 hours after the first [echo] '. This cannot be a valid description of your methods, and a baseline characteristic should be included of age, sex, and at least disopyramide doses”.

Response to comment 1:

Thanks for that comment – on page 4 study protocol it was stated that the first study was done at least 12 hours after the usual dose of disopyramide. “Immediately on completion of the first echocardiographic examination, patients were asked to take their usual morning dose of disopyramide in the presence of the investigators.” We believe this describes exactly the timing of disopyramide administration.
The second echo study was done at least 2.5hrs after the disopyramide administration. Approximately was changed to “at least”. Sex and gender are described in results already in page 6, results 1st paragraph– “Mean age of the remaining 19 patients was 66.1±9.45 years; 57% were men.  Mean time from diagnosis of HOCM was 8.0±9.9 years.” We added also the disopyramide mean dose.

Comment 2:

Results: The average GLS decreased from -17% to -14% after disopyramide in the dose that was given. In particular, GLS was already lower in some basal and septal parts and after disopyramide the baseline GLS decreases in the basal anteroseptal part from -14 to -12% (p= 0,057).

Response to comment 2:

The relation between site of hypertrophy and reduced longitudinal strain has been previously described, this study confirms that observation. This study was powered to detect an absolute reduction of strain from -17 to -14 (3 strain points), please see reply to next reviewer. Smaller reductions are expected when baseline strain is lower and absolute decrease is less than 3 points. In effect, basal strain decreased from -17 to -14 (3 points), while basal anterior and septal strain decreased from -15 to the same -13 (2 points).

Comment 3:

In the abstract and manuscript this anteroseptal part that would be mainly involved in obstruction is converted in a 'basal septal segment involved in left ventricular outflow tract obstruction' but it was the average basal inferior septal segment in your study that was significantly decreased, and to a lesser extent the anteroseptal basal part.

Response to comment 3:

Thanks, I am not sure that assumption is warranted. First, both beta blockers and disopyramide act globally on the myocardium (as seen here too). Secondly, previous studies (Carasso S, et al, Myocardial mechanics explains the time course of benefit for septal ethanol ablation for HCM. J Am Soc Echocardiogr. 2008 May;21(5):493-9. doi: 10.1016/j.echo.2007.08.020) – clearly showed that global myocardial dysfunction reduces LVOT obstruction. The same may happen in the “burnt out” stage of HOCM when LV systolic function decrease. Regional reduction of basal septal obstruction can only be obtained by chemical (scarring) or surgical septal myocardial reduction and enlargement of the tract.

Comment 4:

Possibly, the paired intraindividual results/ changes in anteroseptal GLS would be useful to see, and maybe they correlate with LVOT obstruction. It could be that disopyramide is not useful for everyone, and only in those with higher anteroseptal basal GLS (so that there is something to inhibit)?  

Response to comment 4:

Please see answer to your previous comment

Comment 5:

At present, there are discrepancies that would really prevent this paper from being understood. The conclusion that one would be tempted to follow is that despite GLS reductions, this is not enough GLS reduction to cause a significant effect on LVOT gradients, neither was this reduction enough to cause a significant effect on LVEF.  

The real question is how much GLS decrease is necessary for a LVOT gradient decrease and at which disopyramide dose does the LVOT gradient decrease, and this is not answered in the present study, but is only made difficult to understand.    

Response to comment 5:

In new reference 18 (Carasso at al.) a correlation is shown between GLS and the LVOT gradient. To achieve an LVOT strain below 40mmHg GLS should less than |-13|. For less than 30mmHg the cut-off would be less than |-11|. We added a comment regarding this in the discussion section – page 8. “In a study measuring strain before, during and after septal alcohol ablation18 alcohol was found to acutely reduce global left ventricular strain. GLS was correlated with LVOT gradient, showing that decrease of LVOT gradients to less than 30mmHg would require GLS to be less (negative) than -11%. This reduction was not reached in our current study, maybe because the post administration echo was done before the needed 3-4 hours of plasma peak levels of disopyramide.”

Reviewer 2 Report (New Reviewer)

The Authors of the manuscript found in patients with hypertrophic cardiomyopathy a significant reduction of longitudinal myocardial strain following disopyramide administration. The Authors should address the following issues:

1)      The enrolled study population is too small.

2)      In the Statistics, the Authors should specify if the normal distribution of the data has been assessed to establish the applicability of parametric analysis and how the sample size required by the study has been estimated.

3)      Atrial fibrillation should be mentioned as an exclusion criterion.

4)       LV obstruction mechanism (SAM-related or non-SAM-related) should be reported.

5)      The clinical symptoms in the study group were not reported.

6)      Despite its statistically significant changes, the longitudinal strain did not impact the left ventricular gradient.

Author Response

Reviewer 2 Comments

Comment 1:

The Authors of the manuscript found in patients with hypertrophic cardiomyopathy a significant reduction of longitudinal myocardial strain following disopyramide administration. The Authors should address the following issues:The enrolled study population is too small.

Response to comment 1:

Thank for your suggestion, we added in Page 5: “Sample size assessment was calculated to detect a significant drop in longitudinal strain from 17%±2.5 (SD) (absolute value, typical of HCM patients) of 3% (to 14%) with lowest probability type I (p<0.05) and II (p<0.01) errors and was found to be 14 patients. We prospective recruited 35% more patients for the study.”

Comment 2:

In the Statistics, the Authors should specify if the normal distribution of the data has been assessed to establish the applicability of parametric analysis and how the sample size required by the study has been estimated.
Response to comment 2:

Page 6: “To ascertain normal distribution of pre and post disopyramide strain measurements using the Shapiro-Wilk test.  After confirmation of normality…”

Comment 3:

Atrial fibrillation should be mentioned as an exclusion criterion. 
Response to comment 3:

Page 4: “Patients in atrial fibrillation at the time of echocardiography were also excluded.”

Comment 4:

LV obstruction mechanism (SAM-related or non-SAM-related) should be reported.

Response to comment 4:

All were SAM Related patients with LVOT obstruction. Added to description in results, page 6.

Comment 5:

The clinical symptoms in the study group were not reported.
Response to comment 5:

Thanks for your suggestion, it was Added to results in page 6.

Comment 6:

Despite its statistically significant changes, the longitudinal strain did not impact the left ventricular gradient.

Response to comment 6:

In new reference 18 (Carasso at al.) a correlation is shown between GLS and the LVOT gradient. To achieve an LVOT strain below 40mmHg GLS should less than |-13|. For less than 30mmHg the cut-off would be less than |-11|. We added a comment regarding this in the discussion section – page 8. “In a study measuring strain before, during and after septal alcohol ablation18 alcohol was found to acutely reduce global left ventricular strain. GLS was correlated with LVOT gradient, showing that decrease of LVOT gradients to less than 30mmHg would require GLS to be less (negative) than -11%. This reduction was not reached in our current study, maybe because the post administration echo was done before the needed 3-4 hours of plasma peak levels of disopyramide.”

Round 2

Reviewer 1 Report (Previous Reviewer 2)

The authors have provided the mean disopyramide dose that was given (mean 380 mg and SD 150), and adapted the time to the second echo as at least 2,5 hours after dosing. 

In addition, some insight was given from reference 18 (which is the same as reference 11), about how much GLS reduction would be necessary to see an effect on LVOT obstruction, so as to explain why there was no LVOT obstruction in their patient population. The authors can remove reference 18 and refer to the same reference 11. 

Remaining comments

1. Although you can report that disopyramide decreases GLS, the suggestion in abstract and discussion is that these GLS decreases that are found at the basal septal part 'can explain the mechanism of action of disopyramide in HOCM', which is a suggestion and is not what you found, at least you did not find LVOT gradient decreases. Of course, it would be interesting to know, but that is something else. In the authors reply you are also thinking about global decreases in LV function that would decrease LVOT gradients, not being dependent on specific parts of the ventricle. But also this latter question is not answered in the present study, so you cannot suggest either in the abstract as being responsible for a HOCM effect. You can discuss it, but please do not insert the suggestion in the results of the abstract (or conclusion) or in the conclusion of the manuscript. 

2. My previous suggestion to have paired findings of the results instead of means of the group findings may not have been understood well by the authors, but does not have to be analysed. The idea was to have differences in GLS constructed against baseline findings, but I do not think it would alter the conclusions.   

3. Discussion: "this is the first study to evaluate the acute effect of disopyramide on myocardial mechanics in patients with HOCM".  You probably mean the acute effects of disopyramide on myocardial mechanics as measured with strain, because there have been previous studies on mechanics (ref 7 eg). 

4. Discussion p 10: " Short-acting disopyramide had no effect on other echocardiographic parameters, such as ejection fraction or LVOT gradients. This finding is probably attributable to the short wash-out period of the medication in our chronically disopyramide-treated patients". Some lines further, you also state under limitations: "The brevity of the washout period may have led to an underestimation of the full effect of the drug". In the discussion, nothing is said about the dose of the drug, or about checking the medication levels in the blood, which is a severe limitation of the study. You should acknowledge that it may have been the dose of the drug that may have been too low to be effective on the LVOT gradient. I understand that this is a little awkward to state, but to understand your findings, this has to be inserted in discussion and limitation. You can use the reference of Sherrid et al, in which they report that higher doses (2 x 250 mg instead of 2 x 125 mg) were needed to obtain the best results in lowering LVOT gradients. Sherrid MV, Arabadjian M. A primer of disopyramide treatment of obstructive hypertrophic cardiomyopathy. Prog Cardiovasc Dis. 2012 May-Jun;54(6):483-92. doi: 10.1016/j.pcad.2012.04.003. PMID: 22687589.. 

5. Discussion page 9: "We did not find an effect of disopyramide on global synchrony". I could not find these results in a table or text.  

6. Introduction: the reference 11 was cited by stating that the gradient rebounded when global function improved. Please add 'partially' before rebounded. 

Reviewer 2 Report (New Reviewer)

The Authors have appropriately revised the manuscript.

This manuscript is a resubmission of an earlier submission. The following is a list of the peer review reports and author responses from that submission.

Round 1

Reviewer 1 Report

The present study aims to assess the impact of short-acting disopyramide in patients with hypertrophic obstructive cardiomyopathy (HOCM) using two-dimensional speckle-tracking echocardiography.

They prospectively recruited 19 patients with HOCM patients in chronic treatment with disopyramide and performed an echocardiographic examination after 12 hours from the last disopyramide dose suspension and a second echocardiographic examination at 2.5 hours after disopyramide intake.

After intake of disopyramide, they observed a significant reduction in the mean global strain peak (-17±2% vs -14±2%, p <0.0001), in the basal septal (p=0.015), basal inferior (p=0.019), basal posterior (p=0.05), apical anterior (p=0.0001) and apical lateral segments (p=0.021) strain, a significant accentuation of the base-apex strain gradients (p=0.036).

The authors conclude that strain analysis may be a useful tool to assess the negative inotropic effect of cardiovascular medication on the left ventricle in patients with HOCM.

English is good, the topic of interest although for a restricted target of cardiologists working in the field of cardiomyopathies. Despite the Authors explains that previous studies have failed to demonstrate a negative inotropic effect of the drug because of the small population sizes, the cohort of their study is quite small (19 patients). Statistical analysis is not clear and methods are weak.

The study presents in my opinion a number of methodological issues 

MAJOR COMMENTS:

- METHODS: disopyramide reaches the peak plasmatic concentration 90 minutes after the administration of 200 mg. Why did you perform the second echocardiography y 2.5 hours after the administration?

-METHODS: The kinetic of the drug changes with the renal function. Considering the mean age of  your patients, I would expect some had chronic kidney disease. What are the eGFR and the urinary albumin-to-creatinine ratio? How many had CKD? In which stage?

-METHODS: what was the dosage of disopyramide administered before the second echocardiographic examination?

-METHODS: meals can influence the obstruction gradient: was the echocardiographic examination performed fasting on both occasions?

.METHODS and RESULTS: how was the diagnosis of HOCM performed? Were these patients affected by the inherited disorder? What was the mean wall thickness? What was their functional class NYHA? How many were symptomatic? Were these patients genetically tested? If yes, which genes were tested and how?

-METHODS: strain analysis is affected by a great intra and inter-observer variability in the strain analysis. What was the experience of SC in strain analysis? Who was the other doctor that performed the analysis for the inter-observer variability? Considering the small size, why did not you tested the interobserver variability for all the patients?

-STATISTICAL ANALYSIS: What statistical analysis did you use for the interobserver variability? What statistically analysis did you perform for all the pre- and post- disopyramide echocardiographic examination analyses?

-RESULTS: no details about the population are provided: what was the chronic dose of disopyramide? Which other drugs were the patients on? What was the dose of betablockers? How many gene positive patients? For which genes? How many symptomatic patients? SCD Risk score? Family history? What was the BMI of this patient?

-RESULTS: echocardiographic analysis: no details are provided of the other echocardiographic characteristics, ie. mean wall thickness? How many concentric/asymmetrical hypertrophy? Atrial dimension? Left ventricular volumes? It is well know that the obstruction gradient is also affected by the cavity size…

-DISCUSSION: all the discussion is based on CMR findings although the patients of the present study have not undergone CMR. I would suggest changing the discussion. I would focus on the pharmacological properties of disopyramide and on what the strain analysis consider and why it is suitable for the aim of the study and comparison with other strain analysis studies in HCM, although not focused on obstruction

-DISCUSSION: I would suggest reading the recent published paper by Monda E et al “Bisoprolol for treatment of symptomatic patients with obstructive hypertrophic cardiomyopathy. The BASIC (bisoprolol AS therapy in hypertrophic cardiomyopathy) study” IJC 2022 May 1;354:22-28. doi:10.1016/j.ijcard.2022.03.013

MINOR COMMENT:
List of authors: after the senior author there is an “and”… who is the missing co-author?

Reviewer 2 Report

The manuscript "Impact of short-acting disopyramide on left ventricular mechanics evaluated by strain analysis in patients with hypertrophic obstructive cardiomyopathy" describes the effect of disopyramide on left ventricular strain and systolic rotation in 19 HOCM patients who were previously treated with disopyramide. The results show that disopyramide in the unknown dose given did not reduce the LVOT gradient from baseline levels, but mildly reduced global longitudinal strain. 

The study has design flaws that makes correct interpretation impossible. The medication doses at baseline or for the study were not given, but even then, what medication dose would have been effective in your population? I would have argued that if the medication dose would have decreased LVOT  gradients, this dose would have shown its efficacy. Now we don't know whether the dose given had any effect at all, because there was not even a small reduction in LVOT gradients after giving your dose of dispyramide. This could have been caused by the short wash out period, as you mention in limitation, but even that is unknown. The half time of disopyramide is described as between 4 and 10 hours, so there would have been remaining levels of disopyramide in the blood at baseline. It would have been simple to measure the medication blood levels at baseline. Then, the maximal effect of disopyramide is present at 4-6 hours after dosing, and not at 2,5 hours. Also here, you could have measured the medication levels after this dose. The argument in the literature about how large the effects of disopyramide are, is that at low doses (2x 125 mg) it may not have the required effect, and that at higher doses (2x 250 mg), the best results are obtained (Sherrid MV, Arabadjian M. A primer of disopyramide treatment of obstructive hypertrophic cardiomyopathy. Progr Cardiovasc Dis 2012;54:483-492).  

At present, the study has too severe limitations, despite all its efforts. The study question remains interesting (at least adapted: what does disopyramide do on GLS in humans with HOCM; the study question should not be if disopyramide is a medication with negative inotropic effect, this has been established also in animal preparations such as the one by Kajimoto et al Am J Cardiol 2010,106:1307-1312). The methods should include medication level measurements, and stopping for 3 halve lives of medication before the study starts. I would also do a dose- response study, to see how much GLS reduction is required for a reduction of LVOT gradients. Whether GLS measurement can be used to study whether these medications cause diminished intropy, remains to be seen (GLS is not load-independent measure), that is a reasoning mistake from the conclusions, you use GLS as method but not as study object.